# Smart Conditioning with Venetoclax-Enhanced Sequential FLAMSA + RIC in Patients with High-Risk Myeloid Malignancies

**DOI:** 10.3390/cancers16030532

**Published:** 2024-01-26

**Authors:** Felicitas Schulz, Paul Jäger, Johanna Tischer, Alessia Fraccaroli, Gesine Bug, Andreas Hausmann, Ben-Niklas Baermann, Patrick Tressin, Alexander Hoelscher, Annika Kasprzak, Kathrin Nachtkamp, Johannes Schetelig, Inken Hilgendorf, Ulrich Germing, Sascha Dietrich, Guido Kobbe

**Affiliations:** 1Department for Hematology, Immunology and Clinical Oncology, Heinrich-Heine-University Düsseldorf, 40225 Düsseldorf, Germany; paulsebastian.jaeger@med.uni-duesseldorf.de (P.J.); alexandersebastian.hoelscher@med.uni-duesseldorf.de (A.H.); kobbe@med.uni-duesseldorf.de (G.K.); 2Department of Medicine III, LMU University Hospital, Ludwig-Maximilians-University, 80539 Munich, Germanyalessia.fraccaroli@med.uni-muenchen.de (A.F.); 3Department of Medicine 2, University Hospital, Goethe University Frankfurt, 60590 Frankfurt, Germany; g.bug@em.uni-frankfurt.de; 4Frankfurt Cancer Institute, Goethe University, 60590 Frankfurt, Germany; 5German Cancer Consortium (DKTK), Partner Site Frankfurt/Mainz and German Cancer Research Center (DKFZ), 69120 Heidelberg, Germany; 6Department of Hematology, Oncology, Immunology, Palliative Care, Munich Clinic Schwabing, 80804 Munich, Germany; hausmann.andreas@googlemail.com; 7Medical Clinic I, Department of Hematology, University Hospital Carl Gustav Carus, Technische Universität Dresden, 01062 Dresden, Germany; 8Klinik für Innere Medizin II, Abteilung für Hämatologie und Onkologie, Universitätsklinikum Jena, 07747 Jena, Germany

**Keywords:** FLAMSA, allogeneic transplant (aHSCT), sequential conditioning, AML, MDS, myeloid malignancies, reduced-intensity conditioning, RIC, venetoclax

## Abstract

**Simple Summary:**

Allogeneic stem cell transplantation (aHSCT) is the only potentially curative treatment option for patients with high-risk myeloid malignancies. Depending on the underlying genetic risk profile, up to 50% of patients die of relapse even after aHSCT in first remission. Therefore, further improvement of conditioning regimens is an urgent medical need. Current sequential conditioning regimens combine intensive AML-like induction therapy with total body irradiation or alkylators like treosulfan or melphalan. The first and currently widely accepted prototype of this treatment strategy is the fludarabine/amsacrine/ara-C (FLAMSA) protocol, which has been modified multiple times by changing parts or adding additional cytotoxic drugs to improve clinical results. Knowing that venetoclax, an inhibitor of B-cell lymphoma-2 protein (BCL2), has synergistic effects to chemotherapy without increasing the level of non-hematologic toxicity, several German transplant centers have added venetoclax to the FLAMSA protocol as an individualized treatment approach to improve long term outcome in patients with high-risk myeloid malignancies.

**Abstract:**

Up to 50% of patients with high-risk myeloid malignancies die of relapse after allogeneic stem cell transplantation. Current sequential conditioning regimens like the FLAMSA protocol combine intensive induction therapy with TBI or alkylators. Venetoclax has synergistic effects to chemotherapy. In a retrospective survey among German transplant centers, we identified 61 patients with myeloid malignancies that had received FLAMSA-based sequential conditioning with venetoclax between 2018 and 2022 as an individualized treatment approach. Sixty patients (98%) had active disease at transplant and 74% had genetic high-risk features. Patients received allografts from matched unrelated, matched related, or mismatched donors. Tumor lysis syndrome occurred in two patients but no significant non-hematologic toxicity related to venetoclax was observed. On day +30, 55 patients (90%) were in complete remission. Acute GvHD II°–IV° occurred in 17 (28%) and moderate/severe chronic GvHD in 7 patients (12%). Event-free survival and overall survival were 64% and 80% at 1 year as well as 57% and 75% at 2 years, respectively. The off-label combination of sequential FLAMSA-RIC with venetoclax appears to be safe and highly effective. To further validate these insights and enhance the idea of smart conditioning, a controlled prospective clinical trial was initiated in July 2023.

## 1. Introduction

High-risk myeloid malignancies like (secondary) acute myeloid leukemia (AML), myelodysplastic syndromes (MDS), chronic myelomonocytic leukemia (CMML), and chronic myeloid leukemia (CML) constitute more than 50% of indications for allogeneic hematopoietic stem cell transplantation (aHSCT) in Europe [1,2]. Due to a potent graft-versus-leukemia (GvL) effect [3], aHSCT is the most effective antineoplastic treatment in these diseases and the only potentially curative treatment option. Depending on the individual risk profile, which is defined by underlying genetics, response to induction therapy, or a history of preceding neoplasia and/or chemotherapy [4], relapse rates after aHSCT remain high and up to 50% of patients die of relapse [5]. Therefore, further improvement in conditioning regimens, especially in elderly patients with a potentially high non-relapse mortality (NRM) [6], is an urgent medical need. Current sequential conditioning regimens combine intensive AML-like induction therapy with either myeloablative (MAC) or reduced-intensity conditioning (RIC) using total body irradiation (TBI) or alkylators like busulfan, treosulfan, and melphalan [7,8,9,10]. The first and currently widely accepted prototype of this treatment strategy, which originally combined cytoreductive chemotherapy with reduced-intensity conditioning and prophylactic infusion of donor lymphocytes (DLIs), is the fludarabine/amsacrine/ara-C (FLAMSA) protocol published by Schmid and Kolb in 2005 [4]. The aim was to balance effective eradication of the leukemic clone with a reduction in regimen-related toxicity. Over the years, this protocol has become one of the most often-used conditioning regimens in Europe and has been modified multiple times by changing or adding additional cytotoxic drugs to further improve clinical results [11,12,13,14,15]. Although some protocols were less toxic (e.g., combination of FLAMSA with treosulfan), no study was able to show a higher efficacy [16].

Venetoclax, an inhibitor of B-cell lymphoma-2 protein (BCL2), has been shown to selectively target leukemia stem cells (LSCs) due to their overexpression of BCL2. In addition, animal studies have shown that venetoclax facilitated engraftment and GVL-effect in mice without higher rates of graft-versus-host disease (GVHD) by pharmalogically controlling natural killer cells [17,18]. With the knowledge that venetoclax has synergistic effects to chemotherapy without increasing the level of non-hematologic toxicity as was shown in its combination with azacytidine, decitabine, or FLAG-Ida [19,20], it has already been investigated in combination with RIC conditioning using fludarabine und busulfan [21]. To reduce relapse rates after aHSCT and improve long-term survival/outcomes in patients with high-risk myeloid malignancies, several German transplant centers added venetoclax to one of the well-known FLAMSA protocol variants as an individualized, off-label treatment approach.

## 2. Patients and Methods

### 2.1. Patient

Sixty-one patients (median age 58 years, range 20–74, 29 female) from six different German transplant centers who received FLAMSA-based sequential conditioning in combination with venetoclax between 2018 and 2022 were identified through questionnaires sent to members of the German Cooperative Transplant Study Group and included in this retrospective analysis. The study was approved by the ethics committee of the Heinrich-Heine University Duesseldorf and all patients gave written informed consent for individualized treatment. Patients suffered from myeloid malignancies (40 AML, 18 MDS, 2 CMML, and 1 CML), and 45 (74%) had high-risk genetic features. A total of 60 patients had active disease at time of transplant (19 untreated, 41 relapsed/refractory). For 44 patients, a 10/10 HLA matched donor was available (32 unrelated, 12 related), and 17 patients received allografts from mismatched donors (10 haploidentical, 7 with a 9/10 HLA match). Concomitant diseases were registered using the Hematopoietic Cell Transplantation Comorbidity Index (HCT-CI) [22,23]. At the time of analysis, the median follow-up time was 548 days (range 130–1398) in surviving patients. Detailed patient demographics are shown in Table 1.

### 2.2. Treatment

Patients received fludarabine 30 mg/m^2^, amsacrine 100 mg/m^2^, and cytarabine 2000 mg/m^2^ on four consecutive days (FLAMSA). In patients > 60 years, cytarabine was age-adjusted to a dose of 1000 mg/m^2^. To reduce the risk of a tumor lysis syndrome (TLS), two patients received low-dose cytarabine in an absolute dose of 100 mg per day on three consecutive days before FLAMSA. FLAMSA was then followed by treosulfan 10 g/m^2^ on three consecutive days in 22 patients (36%), and 29 patients (47.5%) received melphalan at an age-adjusted dose of 100–200 mg/m^2^ (200 mg/m^2^ for patients < 50 years, 150 mg/m^2^ for patients aged between 50 and 60 years, and 100 mg/m^2^ for patients > 60 years). Within the groups of FLAMSA + Mel and FLAMSA + Treo, three patients and one patient, respectively, received fludarabin/cyclophosphamid in addition as published in the original protocol by Luznik et al. for patients with a haploidentical donor [24]. Eight patients (15%) were treated with the original FLAMSA+RIC protocol (4 Gy TBI on one day followed by cyclophosphamide 60 mg/kg body weight (BW) on 2 consecutive days) [4]. One patient received a combination of FLAMSA + cyclophosphamide + 8 Gy TBI, while another patient had a combination of FLAMSA + 4 Gy TBI. Five out of ten patients with a haploidentical donor additionally received 2 Gy TBI at the day of transplant. Venetoclax was added one-to-two days before FLAMSA and continued until the first day after FLAMSA or the day before the alkylator or TBI, respectively, at a dose of 20–800 mg daily, with a total dose ranging between 380 and 4700 mg. A ramp-up phase for venetoclax was given to all patients and followed institutional standards. Ramp up starting doses ranged from 20 to 200 mg. Two patients were on continuous venetoclax therapy before conditioning and received a total dose of 6370 mg and 9970 mg. A total of 42 patients (8 related, 34 unrelated donors) received ATG (grafalon) in 3 daily doses between 5 and 20 mg/kg. PtCy was administered to 15 patients (10 haplo, 5 MMUD) on day +3 and +4 after aHSCT at a dose of 50 mg/kg BW per day. Four patients with HLA-identical sibling donors did not receive either ATG or post-transplant cyclophosphamide. Immunosuppressive therapy following allografting consisted of a combination of tacrolimus and mycophenolate mofetil or cyclosporine A and mycophenolate mofetil according to institutional standards. Supportive care, like the application of granulocyte-colony stimulating factor to hasten neutrophil recovery and the administration of antimicrobial prophylaxis also followed institutional standards.

### 2.3. Monitoring and Definitions 

After discharge from the hospital, outpatient and subsequent treatment as well as the time of response evaluation followed institutional standards. Complete remission was defined as < 5% blasts in the bone marrow (BM) by cytomorphology, no evidence of dysplasia, and complete count recovery with platelet count > 100.000/µL and absolute neutrophil count > 1000/µL. Time of leukocyte engraftment was defined as the first of 3 consecutive days with a white blood cell count > 1000/µL. Concordantly, time of platelet engraftment was equivalent the first of 3 consecutive days with platelet counts > 20.000/µL. Grading of acute and chronic graft-versus-host disease (aGVHD, cGVHD) was performed according to the established criteria [25,26].

### 2.4. Statistics 

Event-free survival (EFS), relapse-free survival (RFS), overall survival (OS), non-relapse mortality (NRM), as well as treatment response and GVHD onset were calculated from the day of transplantation to the respective event. Death—independent of its cause—was rated as an event in the case of OS. Death and relapse were both separate events with regard to EFS. All time-to-event curves were estimated using the Kaplan–Meier method [27]. Statistical analyses were performed using IBM SPSS statistical software (Version 27). 

## 3. Results

### 3.1. Hematologic Reconstitution and Chimerism 

All evaluable patients were successfully engrafted and hematologic reconstitution of white blood cells (WBC) > 1000/µL occurred after a median of 15 days (range 8–41 days). Median time to platelet recovery was 19 days (range 2–78 days) with four patients (7%) never reaching > 20,000 platelets/µL. Analysing hematopoietic reconstitution exclusively for those patients who did not receive ptCy, engraftment of WBC occurred after a median time of 13 days, while median time to platelet recovery was 17 days in these patients. At day +30, 52 patients (87%) had complete donor chimerism, while 4 patients (7%) had a donor chimerism < 90%. Three months after transplant, 49 patients (83%) had complete donor chimerism while 5 patients (9%) had a donor chimerism < 90%.

### 3.2. Toxicity and Infections 

Tumor lysis syndrome (TLS), as the most dreaded side effect during the combination of FLAMSA and venetoclax, occurred in only two patients (3%). Toxicities and complications during inpatient stay were mainly infections as listed in Table 2. Veno-occlusive disease (VOD), a possibly life-threatening complication in transplant setting, occurred in one patient (1.6%) and was successfully treated with defibrotide. No significant extra-hematologic toxicity possibly related to venetoclax was observed.

### 3.3. Acute and Chronic GVHD 

Acute GVHD occurred in 37 patients (61%), with 24 patients (39%) developing grade I to II aGVHD and 13 patients (21%) developing grade III–IV aGVHD, with only one patient having grade IV GVHD. According to the National Institutes of Health (NIH) criteria, 24 patients (39%) had classic aGVHD and 12 patients (20%) had late-onset aGVHD. According to the NIH criteria for chronic GVHD, 16 patients (26%) had cGVHD, with 11 patients (18%) having mild to moderate and 5 patients (8%) suffering from severe cGVHD. Treatment of GVHD followed institutional standards. Rates of aGVHD and cGvHD are shown in Figure 1.

### 3.4. Disease Response, Relapse, and Survival

At day +30 after aHSCT, 25 patients (41%) achieved a complete hematological remission. Thirty patients (50%) had complete hematological remission with incomplete recovery (CRi), i.e., meaning a platelet count <100 × 10^9^/L and/or an absolute neutrophil count <1 × 10^9^/L resulting in a CR/CRi rate of 91%. Five patients (8%) had ongoing active disease and one patient was not evaluable due to early death in aplasia on day +15 from septic shock. After a median follow-up of 548 days, 44 patients (72%) were alive (38 patients in ongoing remission, 8 patients after salvage therapy, and 6 patients in relapse receiving further treatment). Twenty-three patients (38%) experienced relapse after a median of 172 days (range 62–943). Of note, 5 of these patients (22% of relapsed patients) had extramedullary AML manifestations without bone marrow involvement. Of the 5 patients who had active disease at day +30 after aHSCT, 2 patients were alive in complete remission (one after accelerated reduction in immunosuppressive therapy, hereby developing aGVHD, and the other one after salvage therapy with hypomethylating agents plus venetoclax and DLIs). The remaining 3 patients died from their disease. Seven patients (12%) received DLIs in the context of salvage chemotherapy, which led to a second, ongoing remission after aHSCT in 3 patients. Another 5 patients (8%) underwent secondary aHSCT after relapse, of whom 3 patients were still alive at the time of analysis. There was a statistically significant difference regarding cumulative incidence of relapse between pre-treated patients and those who underwent upfront aHSCT (*p* = 0.014) as shown in Figure 2. Relapse occurred less frequently and later in time in patients that had not received any antineoplastic treatment before conditioning. Among all patients, overall and event-free survival were 80% and 64% at 1 year as well as 75% and 57% at 2 years, respectively (Figure 3). Classifying patients into two groups regarding the type of donor (10/10 match vs. mismatched donor including haplo and MMUD 9/10), the difference in OS was of borderline significance in favor of 10/10 matched donors (*p* = 0.09). Dosing of venetoclax did not influence response. Regarding the outcome of TP53-mutated patients (six patients had a double-hit mutation and four patients a single-hit), all patients with double-hit status died (5 because of relapse/active disease und one patient due to infection). All patients with a single-hit TP53-mutation were alive and in complete remission. Treatment-related mortality (TRM) was 2% at 3 months and 7% at 1 year. Cumulative incidence of TRM is shown in Figure 4. In total, 17 patients (28%) died. Of these, 12 patients (20%) died because of relapse, 2 patients (3%) had lethal infections, and in 3 patients (5%) death was related to infections in the context of severe GVHD.

## 4. Discussion

The results of this multi-center, retrospective analysis suggest that the combination of sequential FLAMSA + venetoclax in a RIC setting may be a new treatment option for patients with high-risk myeloid malignancies and active disease. In light of the older patient populations, TRM is one of the most critical aspects that needs to be considered when deciding which conditioning regimen might be most suitable for each individual patient. Therefore, escalation of dose intensity is not possible in elderly or comorbid patients. The BCL-2 inhibitor venetoclax has recently become an essential element of AML treatment, especially in elderly patients considered unfit for intensive therapy. Following approval in combination with hypomethylating agents, a large amount of data on toxicity and efficacy have been published. Venetoclax has been shown to be highly effective in combination with hypomethylating agents or FLAG-Ida [19,20]. In addition, venetoclax demonstrated increased donor engraftment in the setting of RIC while avoiding the toxicity and GVHD-priming effects of MAC in a mouse model [28]. Furthermore, venetoclax was able to suppress recipient NK cell function via inhibition of BCL2 in mice, thereby avoiding graft rejection without the toxicities associated with more intensive conditioning, including GVHD [18]. The aim of combining FLAMSA with venetoclax was to increase the efficacy of RIC to the level of MAC regimens without generating MAC-equivalent toxicity.

With a median age of 58 years (range 20–74), our patient cohort was older as compared to most earlier publications on the FLAMSA regimen [7,9,29]. Furthermore, with one-third of patients having an HCT-CI ≥ 3, our study population had significant underlying health conditions. In this retrospective analysis, we demonstrate good tolerability even in older patients with relatively low TRM (7% at 1 year) compared to prior analyses such as, for example, FLAMSA in combination with melphalan in patients with high-risk myeloid malignancies who underwent upfront aHSCT [13]. There was only one early death from sepsis. Hematopoietic reconstitution occurred after a median of 15 days regarding white blood cell count and 19 days regarding platelets, which is in line with previous studies using FLAMSA-RIC [12,30,31]. At day +30 after aHSCT, we observed a cumulative CR rate of 91%, which is again in line with response rates of FLAMSA-based conditioning in the literature [32,33]. There was no difference compared to previous FLAMSA-RIC studies regarding the cumulative incidence of acute and chronic GVHD occurring in 61% and 26%, respectively [8,14,34].

The relapse rate remained significant, especially in relapsed/refractory disease, with 23 patients (38%) experiencing relapse after a median of 172 days. This is comparable to results of other studies using FLAMSA-based conditioning regimens. In the context of active disease (AD), relapse rates in our study are lower than those published by Schneidawind et al. or Pfrepper et al. They treated patients with R/R AML in AD with FLAMSA + RIC and documented relapse rates of 52% and 69%, respectively [35,36]. Patients experiencing relapse could be subdivided into early (<12 months post-aHCST) and late relapses (>12 months after aHSCT), with 14 out of 23 relapses (61%) occurring within the first year. Eighty-seven percent of these patients with early relapse had either a complex karyotype or unfavorable disease risk at diagnosis according to molecular analyses. Regarding those patients who underwent upfront aHSCT, only 3 out of 19 patients (16%) experienced relapse so far. All were late relapses as shown in Figure 2. These observations are in line with the analyses of Platte et al., who found cytogenetics, disease risk stratification at diagnosis, and pretransplant strategies to be major determinants for the time of relapse. [37]. Our data regarding timely reconstitution of neutrophils and platelets, rate of acute and chronic GVHD, no additional extra hematologic toxicity, as well as early response and relapse are in line with the results of Garcia et al., who added venetoclax to fludarabin/busulfan and evaluated the combination as feasible and safe [21]. There are further plans to investigate the combination of conditioning regimens and venetoclax in the future. In Melbourne for example, a phase I study in patients with different hematologic malignancies assessing the combination of venetoclax and fludarabine/cyclophosphamide (NCT05005299) is planned, while there will be a randomized phase II study assessing the combination of venetoclax and sequential busulfan, cladribine, and fludarabine in patients with AML or MRD (NCT 02250937) at the MD Anderson Center [38,39]. 

With respect to limitations in our analysis, its retrospective aspect as well as the underlying heterogeneous collective of patients from six different transplant centers in Germany need to be mentioned. Patients received different types of conditioning regimens with different doses of venetoclax; therefore, our analyses are not based on standardized therapies. In addition, supportive care, for example like administration of antimicrobial prophylaxis or immunosuppressive therapy, followed local institutional standards as well.

To objectify and further validate our insights, a controlled prospective trial to evaluate the maximum tolerated dose of venetoclax in combination with FLAMSA will be the next step. If these early results can be confirmed in a phase I/II study, a randomized controlled trial testing FLAMSA + venetoclax compared to the standard of care is needed. 

## 5. Conclusions

The combination of sequential FLAMSA + RIC with venetoclax may be a smart way to extend the limited therapy options for patients with high-risk myeloid malignancies as it appears to be safe and highly effective without increasing the rate of non-hematologic toxicity. To further validate these insights and enhance the idea of smart conditioning, a controlled prospective clinical trial should be the next step.

## Figures and Tables

**Figure 1 cancers-16-00532-f001:**
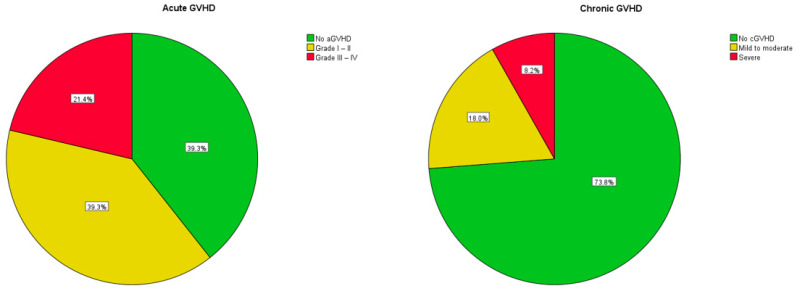
Rates of acute and chronic GVHD.

**Figure 2 cancers-16-00532-f002:**
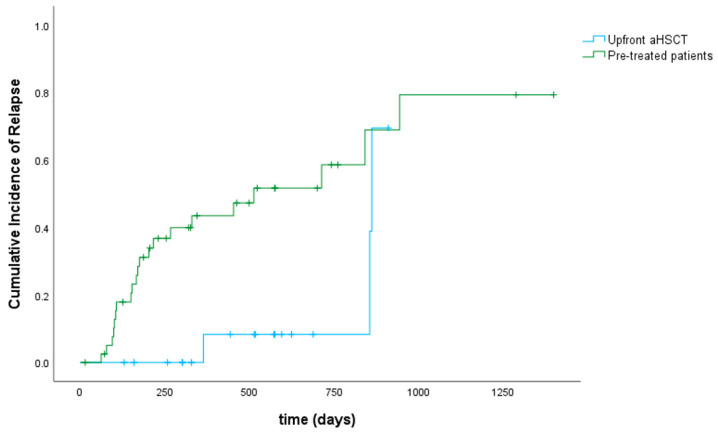
Cumulative incidence of relapse.

**Figure 3 cancers-16-00532-f003:**
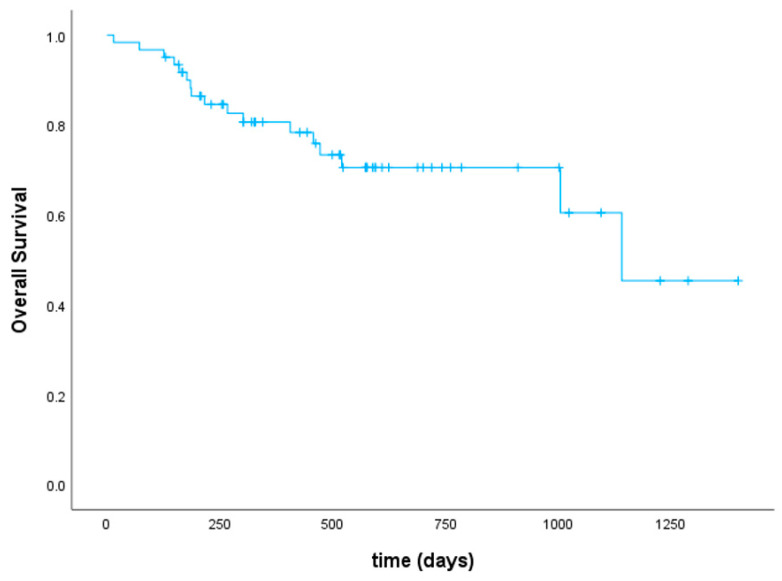
Overall survival.

**Figure 4 cancers-16-00532-f004:**
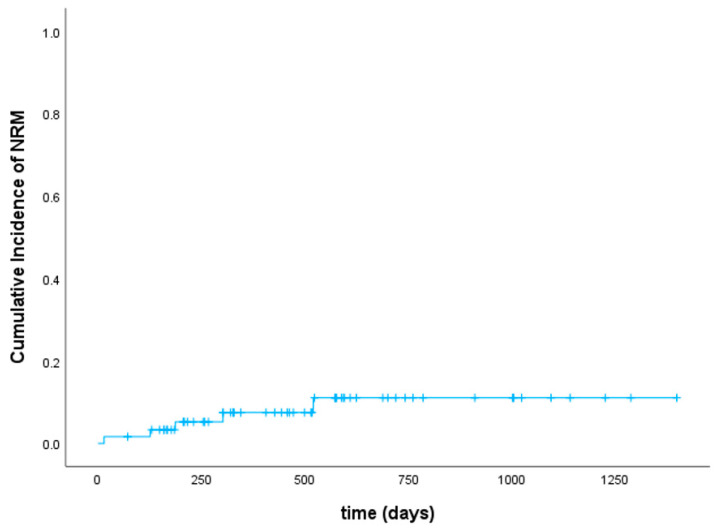
Cumulative incidence of NRM.

**Table 1 cancers-16-00532-t001:** Patient demographics.

Age in years, median (range)	58 (20–74)
Sex, no. (%)	
Male	32 (52.5)
Female	29 (47.5)
Diagnosis, no. (%) AML (according to WHO 2022) with defining genetic abnormalities - AML, myelodysplasia-related - AML with NPM1 mutation - AML with RUNX1::RUNX1T1 fusion - AML with CEBPA mutation - AML with KMT2A rearrangement - AML with CBFB::MYH11 fusion myeloid neoplasm post-cytotoxic therapy defined by differentiation - acute myelomonocytic leukemia - AML with minimal differentiation - acute basophilic leukemia - acute monocytic leukemia	40 (65.6) 31 (50.8) 17 (27.9) 7 (11.5) 3 (4.9) 2 (3.3) 1 (1.6) 1 (1.6) 2 (3.3) 7 (11.5) 3 (4.9) 2 (3.3) 1 (1.6) 1 (1.6)
MDS (according to WHO 2022) with bi-allelic TP53 inactivation with increased blasts 1 with increased blasts 2 MDS with low blasts	18 (29.5) 6 (9.8) 6 (9.8) 5 (8.2) 1 (1.6)
CMML CMML0 CMML1	2 (3.3) 1 (1.6) 1 (1.6)
CML (in blast crisis)	1 (1.6)
Risk stratification according to genetics, no. (%)	
High-risk	45 (73.8)
Intermediate	12 (19.7)
Favorable	4 (6.6)
HCT-CI, no. (%)	
0	21 (34.4)
1–2	19 (31.1)
≥3	19 (31.1)
Unknown	2 (3.3)
Disease status at transplant, no. (%)	
Relapsed/refractory	41 (67.2)
Untreated disease	19 (31.1)
CR	1 (1.6)
Number of transplant(s), no. (%)	
First	58 (95.1)
Second	3 (4.9)
Conditioning regimen, no. (%)	
FLAMSA + melphalan (100–200 mg/m^2^)	29 (47.5)
FLAMSA + treosulfan (30 g/m^2^)	22 (36.1)
FLAMSA + Cy 60 mg/kg + TBI 4 Gy	9 (14.8)
FLAMSA + Cy 60 mg/kg + TBI 8 Gy	1 (1.6)
FLAMSA + TBI 4 Gy	1 (1.6)
Donor, no. (%)	
Matched unrelated donor (MUD)	32 (52.5)
Matched related donor (MRD)	12 (19.7)
Haploidentical	10 (16.4)
Mismatched unrelated donor 9/10 (MMUD)	7 (11.5)
GVHD prophylaxis, no. (%)	
Antithymocyte globulin 3 × 5 mg/kg BW	5 (8.2)
Antithymocyte globulin 3 × 10 mg/kg BW	30 (49.2)
Antithymocyte globulin 3 × 20 mg/kg BW	7 (11.5)
Post-Transplant cyclophosphamide 50 mg/kg BW	15 (24.6)
None	4 (6.6)

**Table 2 cancers-16-00532-t002:** Toxicities and complications.

	No. (%)
Infectious	
● Viral	
○ EBV reactivation ○ CMV reactivation ○ HHV6 encephalitis ○ HSV stomatitis ○ HSV esophagitis ○ BKV cystitis ○ Hepatitis C-reactivation	4 (6.6) 2 (3.3) 1 (1.6) 1 (1.6) 1 (1.6) 1 (1.6) 1 (1.6)
● Bacterial ○ Sepsis ○ Bacteremia ○ Erysipel ○ Toxoplasmosis	4 (6.6) 3 (4.9) 1 (1.6) 1 (1.6)
● Fungal ○ Pneumonia	5 (8.2)
● Other ○ Fever of unknown origin ○ Mucositis ○ Pneumonia ○ Pleuritis ○ Colitis	2 (3.3) 2 (3.3) 1 (1.6) 1 (1.6) 1 (1.6)
Cardiac ● Atrial fibrillation ● Sinus tachycardia ● Refractory arterial hypertension	3 (4.9) 1 (1.6) 1 (1.6)
Hemorrhagic ● Upper gastrointestinal bleeding ● Vitreous hemorrhage	1 (1.6) 1 (1.6)
Other ● Tumor lysis syndrome ● VOD ● Renal insufficiency	2 (3.3) 1 (1.6) 1 (1.6)

## Data Availability

The datasets used and analyzed during the current study are available from the corresponding author on reasonable request. The data are not publicly available due to ethical restrictions.

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
