# Peer review of "Smart Conditioning with Venetoclax-Enhanced Sequential FLAMSA + RIC in Patients with High-Risk Myeloid Malignancies"

_cancers, 2024, doi:10.3390/cancers16030532_

Round 1

Reviewer 1 Report

Comments and Suggestions for Authors

ivThe study of Schulz et al. describes a pilot study for the off label addition of the BCL-2 inhibitor venetoclax to sequential conditioning regimen to allogenic transplantation for patients with high risk myeloid malignancies (mainly AML patients). The majority of patients had active disease. Outcome is promising with 90% of patients obtaining complete remission and event-free survival and overall surval were 64% and 80% at 1 year as well as 57% and 75% at 2 years, respectively. No increased toxicity was observed. The study is interesting and seems to report for the first time addition of venetoclax to sequential conditioning regimens in this patient setting.

Some issues need to be addressed:

1.     Table 1 needs to be revised according to WHO 2022 and ICC classification. It would be very helpful to include genetic characteristics of patients. In particular, information for the outcome of TP53 double hit patients would be very helpful.

2.     The exact scheme of addition of venetoclax to FLAMSA is confusing in the text and should be clarified in a figure.

3.     Was there a ramp up for venetoclax? What were the criteria for dosing as doses ranged from 20 mg to 800 mg daily? It should be discussed whether dosing of venetoclax may have impact or not on response.

4.     A clear statement for off label use of venetoclax would be appreciated.

Reviewer 2 Report

Comments and Suggestions for Authors

Lucid presentation of difficult patient cohort. 

Author Response

First, we would like to thank all reviewers for their constructive and thoughtful comments and suggestions. We are very pleased about the general assessment that our manuscript could be suitable for publication Cancers.

Reviewer 2: Lucid presentation of difficult patient cohort. 

Response to Reviewer: Thank you very much for reviewing our manuscript and your positive feedback!

Reviewer 3 Report

Comments and Suggestions for Authors

Manuscript by Schultz F and Kobbe G, entitled : " Smart conditioning with Venetoclax enhanced sequential FLAMSA + RIC in patients with high-risk myeloid malignancies."

The article focuses on the improvement of treatment protocol for Acute Myeloid Leukemia. Allogeneic transplantation may yield the best treatment for some of the patients suffering life-threatening leukemias. However, treatment is harsh, and not all patients gain the best benefit. This study describes a significant advancement, using the additional BCL2-inhibitor Venetoclax with the broadly used FLAMSA protocol, for some of the patients that are rationally selected.

The paper efficiently describes the data of German transplant centers. It is well-written and very good to read. Data are well presented. Discussion thoroughly connects findings with previous and ongoing clinical experiments, and authors further suggest the fundamental mechanism of drugs that are to be considered per patient. Limitations are adequately considered. Conclusions are sharp.

I find the manuscript very interesting and important for all transplantation experts. It shall be published FAST.

My only possible comment, totally to the consideration of authors, is the use of the word "smart". It might be a lingual preference, and one may consider using "rational" instead. Smart is also fine – the manuscript must be published as soon as possible.

Thanks for putting together this important paper- well done! 

Author Response

First, we would like to thank all reviewers for their constructive and thoughtful comments and suggestions. We are very pleased about the general assessment that our manuscript could be suitable for publication Cancers.

Reviewer 3: The article focuses on the improvement of treatment protocol for Acute Myeloid Leukemia. Allogeneic transplantation may yield the best treatment for some of the patients suffering life-threatening leukemias. However, treatment is harsh, and not all patients gain the best benefit. This study describes a significant advancement, using the additional BCL2-inhibitor venetoclax with the broadly used FLAMSA protocol, for some of the patients that are rationally selected.

The paper efficiently describes the data of German transplant centers. It is well-written and very good to read. Data are well presented. Discussion thoroughly connects findings with previous and ongoing clinical experiments, and authors further suggest the fundamental mechanism of drugs that are to be considered per patient. Limitations are adequately considered. Conclusions are sharp.

I find the manuscript very interesting and important for all transplantation experts. It shall be published FAST.

My only possible comment, totally to the consideration of authors, is the use of the word "smart". It might be a lingual preference, and one may consider using "rational" instead. Smart is also fine – the manuscript must be published as soon as possible.

Thanks for putting together this important paper- well done! 

Response to Reviewer: Thank you very much for reviewing our manuscript and your positive feedback! We would prefer to maintain the word ‘smart’ instead of ‘rational’.